# Self-help and mutual assistance in the aftermath of a tsunami: How individual factors contribute to resolving difficulties

**Motoaki Sugiura**[1,2,3]*, **Ryo Ishibashi**[2,3], **Tsuneyuki Abe**[4], **Rui Nouchi**[2,3], **Akio Honda**[5], **Shosuke Sato**[1], **Toshiaki Muramoto**[1], **Fumihiko Imamura**[1]

**1** International Research Institute of Disaster Science, Tohoku University, Sendai, Japan, **2** Institute of Development, Aging and Cancer, Tohoku University, Sendai, Japan, **3** Smart-Aging Research Center, Tohoku University, Sendai, Japan, **4** Graduate School of Arts and Letters, Tohoku University, Sendai, Japan, **5** Faculty of Informatics, Shizuoka Institute of Science and Technology, Fukuroi, Japan

* sugiura@tohoku.ac.jp

**Data Availability Statement:** All relevant data are within the manuscript and its Supporting Information files.

## Abstract

Self-aid and mutual assistance among victims are critical for resolving difficulties in the immediate aftermath of a disaster, but individual facilitative factors for such resolution processes are poorly understood. To identify such individual factors in the background (i.e., disaster damage and demographic) and personality domains considering different types of difficulty and resolution, we analyzed survey data collected in the 3-year aftermath of the 2011 Great East Japan Earthquake and Tsunami. We first identified major types of difficulty using a cluster analysis of 18 difficulty domains and then explored individual factors that facilitated six types of resolution (self-help, request for help, help from family, help from an acquaintance, help through cooperation, and public assistance) of these difficulty types. We identified general life difficulties and medico-psychological difficulties as two broad types of difficulty; disaster damage contributed to both types, while some personality factors (e.g., neuroticism) exacerbated the latter. Disaster damage hampered self-resolution and forced a reliance on resolution through cooperation or public assistance. On the other hand, some demographic factors, such as being young and living in a three-generation household, facilitated resolution thorough the family. Several personality factors facilitated different types of resolution, primarily of general life difficulties; the problem-solving factor facilitated self-resolution, altruism, or stubbornness resolutions through requests, leadership resolution through acquaintance, and emotion-regulation resolution through public assistance. Our findings are the first to demonstrate the involvement of different individual, particularly personality, factors in survival in the complex social dynamics of this disaster stage. They may contribute to disaster risk mitigation, allowing sophisticated risk evaluation and community resilience building.

**Funding:** M.S. was supported by Special Project Researches (http://irides.tohoku.ac.jp/topics_project/index.html) (H24-A-5 and H25-A-4 to MS) from International Research Institute of Disaster Science, Tohoku University, and Topic-Setting Program to Advance Cutting-Edge Humanities and Social Sciences Research and KAKENHI 17H06219 from Japan Society for the Promotion of Science (https://www.jsps.go.jp/english/index.html). The funders had no role in study design, data collection and analysis, decision to publish, or preparation of the manuscript.

**Competing interests:** The authors have declared that no competing interests exist.

## Introduction

Victims of natural disasters experience a variety of difficulties in the immediate aftermath, such as access to food, water, sanitation, and medical assistance, as well as psychological problems [1–3]. Among the stages of the disaster management cycle (i.e., mitigation, preparation, emergency, and recovery stages), the emergency stage is a critical period for intervention by organizations [4], but such public support takes time and is often limited. Self-aid and mutual assistance among victims in resolving such difficulties are critical for survival before public support becomes available [5–7]. Understanding the individual factors that contribute to successfully resolving difficulties encountered is important for risk evaluation and for fostering citizens' resilience during times of quiescence, namely, the mitigation stage [4]. The emphasis on this aspect is in line with the promotion of a people-centered, preventive approach to disaster risk management, as encouraged in the Sendai Framework 2015–2030 [8].

Published research on such individual factors has primarily addressed factors that contribute to negative consequences, that is, exacerbating factors. In this regard, the social science and mental health literatures have provided rather different images of victims. In the social science literature, the image of victims is rather vague in terms of the difficulties they encounter. Proposals of vulnerability factors [9–11] have been based on reviews of studies that describe vulnerability in association with various outcomes throughout the disaster stages, including casualty, physical or mental health, post-disaster socioeconomic states, and disaster preparedness. They commonly point out low socioeconomic status as a significant vulnerability factor. On the other hand, the image of the victim is clear in the mental health literature, primarily addressing poor mental health outcomes, such as post-traumatic stress disorders (PTSD). Although various individual factors have also been identified as risk factors, robust examination of disaster exposure and psychosocial resources (e.g., psychiatric pre-morbidity and personality) seems to be unique to the mental health literature [12–14]. In both literatures, the contributions of several common individual factors such as age, sex, and education are controversial.

To explore the individual factors that contribute to the resolution of difficulties, the following three issues should be addressed. First, different types of difficulty may be dealt with separately given the apparently distinct characteristics of victims as represented by the social science and mental health fields. We were curious as to whether the distinction reflects a lack of integrated research between the two fields [15] or whether there are indeed different types of difficulty. Second, different types of resolution should be addressed separately, as individual factors likely affect the availability of different resolution types. For example, self-help may be limited by disaster damage, mutual assistance between familiar people may depend on existing social capital, and that between unfamiliar people may require an explicit request or proposal [16,17]. Finally, in addition to common background factors, we were interested in personality traits as individual factors. Understanding personality traits, or individual psycho–behavioral characteristics, may be a promising approach for educational or cultural investigations into improving resilience [7,11].

In this study, we investigated the individual background and personality factors that facilitate self-aid or mutual assistance in the immediate aftermath of a disaster considering different types of difficulty and resolution. We used survey data from the 2011 Great East Japan Earthquake and Tsunami, which included answers to questions about whether survivors experienced and could resolve difficulties in the aftermath of the tsunami across 18 areas (e.g., access to food, clothes, medicine, and psychological aid). When asking about resolution success, we identified six types of approach to resolving each difficulty: self-resolution (i.e., through one's own effort), four types of mutual assistance (i.e., resolution through requests for help,

resolution through family, resolution through acquaintances, and resolution through cooperation), and resolution through public assistance. The dataset also included data on the background (i.e., damage and demographic) and personality factors of each survivor.

First, we explored the major types of difficulty among the 18 areas using a cluster analysis; characteristics of these difficulty types were examined in terms of frequency, relevant resolution types, and exacerbating factors. Then we explored the individual factors that facilitated the resolution of each difficulty type separately for the six solution types. To examine contributing individual factors, a hierarchical regression analysis was used. We first identified contributing background factors and then asked whether any personality factor had additional explanatory power. We addressed two sets of personality factors: one set specifically relevant to survival and the other set for the Big Five personality dimensions. The first is a comprehensive set of psychological and behavioral characteristics known as "power to live" that has been identified as having been advantageous for survival during the 2011 Great East Japan Earthquake and Tsunami [18]. We expected that its different factors facilitated different types of resolution. Of the Big Five personality dimensions, neuroticism is expected to contribute as an exacerbating factor in mental health outcomes [12–14].

## Materials and methods

### Survey

Participants were survivors of the 2011 Great East Japan Earthquake and Tsunami [18]. In early December 2013, a questionnaire battery was mailed to 3,600 residents who were randomly sampled from the electoral registers (and thus were aged 20 years or older) of tsunami-affected districts or temporary settlements in the four most populated coastal cities (Ishino-maki, Kesen-numa, Natori, and Sendai) in Miyagi Prefecture, where the damage caused by the earthquake and tsunami was most severe. In total, 1,412 questionnaires (39%) were anonymously completed and returned by mid-January 2014. The damage and demographic profiles of respondents are shown in Table 1. The survey was approved by the Ethics Committee for Surveys and Experiments of the Graduate School of Arts and Letters at Tohoku University (2012-1019-190749) and was conducted with support from Survey Research Center Co., Ltd. (Tokyo, Japan). See [18] for further details. The data analyzed in this study partially overlap with those that have previously been used to construct a "power to live" inventory and analyze individual factors of self-protectiveness and other supportive behaviors during evacuation to avoid a tsunami [18–21].

### Variables

The areas of difficulty in the aftermath of the earthquake and tsunami were selected to cover as many difficulties as possible based on our own experience and knowledge. The following items were selected: 1) eating, 2) cooking, 3) appropriate spare clothes, 4) room temperature, 5) sleeping, 6) access to a toilet, 7) washing one's face, 8) bathing, 9) laundry, 10) information gathering, 11) transportation, 12) medical care for oneself, 13) medical care for one's family, 14) psychological stress, 15) psychological care for one's family, 16) noise, 17) stench, and 18) privacy. We asked respondents to report experiences: "During the period from the occurrence of the earthquake, tsunami, or evacuation until the restart of a normal life or entering temporary housing." For each item, participants were asked to evaluate the degree to which they experienced difficulty on a scale of 0 to 2, where 0 = none, 1 = a little, and 2 = very much. This was meant to encourage responders with a moderate degree of difficulty to select 1, as they would likely select "none" on a binary scale to differentiate their losses from those of seriously damaged victims. When respondents reported having experienced difficulty (i.e., when they

**Table 1. Damage and demographic profiles of the participants.**

| Damage | | |
|---|---|---|
| Home building | No damage | 137 (10%) |
| | Flood below floor | 49 (4%) |
| | Flood above floor | 31 (2%) |
| | Partially damaged | 175 (13%) |
| | Partially destroyed | 72 (5%) |
| | Largely destroyed | 255 (18%) |
| | Completely destroyed | 681 (49%) |
| Household goods | No damage | 140 (10%), |
| | Partial damage | 304 (22%) |
| | About half damaged | 265 (19%) |
| | Almost all damaged | 693 (49%) |
| Car | Lost all usable cars | 483 (35%) |
| | There was a usable car | 820 (59%) |
| | Did not own a car | 89 (6%) |
| Own injury | No injury | 1,282 (92%) |
| | Injured, but not badly enough to go to the hospital | 74 (5%) |
| | Injured badly enough to go to the hospital | 30 (2%) |
| | Injured badly enough to be hospitalized | 10 (1%) |
| Family injury or death | Injured, but not badly enough to go to the hospital | 76 (6%) |
| | Injured badly enough to go to the hospital | 33 (2%) |
| | Injured badly enough to be hospitalized | 20 (1%) |
| | Died due to the earthquake or tsunami | 119 (9%) |
| | None applies | 1,130 (83%) |
| Friend injury or death | Injured, but not badly enough to go to the hospital | 72 (5%) |
| | Injured badly enough to go to the hospital | 44 (3%) |
| | Injured badly enough to be hospitalized | 35 (3%) |
| | Died due to the earthquake or tsunami | 538 (39%) |
| | None applies | 776 (56%) |
| Refugee life | Yes (experienced) | 911 (67%) |
| | No | 451 (33%) |
| **Demographics** | | |
| Sex | Male | 564 (40%) |
| | Female | 832 (60%) |
| Age | 20s | 87 (6%) |
| | 30s | 144 (10%) |
| | 40s | 216 (15%) |
| | 50s | 274 (20%) |
| | 60s | 391 (28%) |
| | 70s | 280 (20%) |
| | 80 years or older | 5 (0%) |
| Education | Junior high school | 221 (16%) |
| | High school | 738 (53%) |
| | College or junior college | 251 (18%) |
| | University or graduate school | 146 (11%) |
| | Other | 28 (2%) |

(*Continued*)

**Table 1.** (Continued)

| | | |
|---|---|---|
| Occupation (before the earthquake) | Self-employed (including a farmer, forester, fishermen, or family worker) | 182 (13%) |
| | Business owner | 84 (6%) |
| | Worker (including manager) | 550 (40%) |
| | Housework | 127 (9%) |
| | Student | 32 (2%) |
| | Unemployed | 255 (19%) |
| | Other | 130 (10%) |
| Household income (before the earthquake) | Less than 2.00 million JPY | 297 (12%) |
| | 2.00–3.99 million JPY | 420 (32%) |
| | 4.00–5.99 million JPY | 282 (22%) |
| | 6.00–7.99 million JPY | 141 (11%) |
| | 8.00–9.99 million JPY | 80 (6%) |
| | 10.00–11.99 million JPY | 34 (3%) |
| | 12.00–13.99 million JPY | 21 (2%) |
| | 14.00 million JPY or more | 33 (3%) |
| Care recipient (before the earthquake) | Yes (in the family) | 183 (13%) |
| | No | 1173 (87%) |
| Residence mode (before the earthquake) | Own house | 1111 (80%) |
| | Company housing or official residence | 15 (1%) |
| | Public housing | 14 (1%) |
| | Family, friend, or acquaintance's house | 43 (3%) |
| | Private rental housing | 204 (15%) |
| | Other | 2 (0%) |

Data are given as frequency (effective %).

selected 1 or 2), they were also asked to report whether and how the problem was resolved by choosing one or more of the following seven options: 1) by one's own effort (self), 2) by asking someone else for help (request), 3) with the help of family members or relatives (family), 4) with the help of acquaintances (acquaintances), 5) through the mutual cooperation of refugees (cooperation), 6) through public support (municipality, military, or volunteers) (public), and 7) was not resolved (unsolved). The basic statistics of these variables are given in S1 Table in S1 Results.

The background factors include damage and demographic factors (Table 1). The damage factors include the degree of damage to one's home building (home building; 7 levels), the degree of damage to one's household goods (household goods; 4 levels), loss or availability of a car for the household (car; three options: lost, available, or not owned), the degree of one's own injury (own injury; 4 levels), the degree of injury or death of family members (family injury or death; multiple choice from 3 levels in injury and death) and friends (friend injury or death; multiple choice from 3 levels in injury and death), and experience of refugee life (refugee life; yes/no). Demographic factors include sex, age (7 levels/decades), educational background (education; 5 levels), occupation (seven options: self-employed (including a farmer, forester, fishermen, and family worker), business owner, worker (including being a manager), housework, student, unemployed, and other), household income (8 levels/200 million JPY), household structure (five options: single, couple, two generations (parents and children), three

generations (parents, children, and grandchildren), and other), the need of a family member for daily care or assistance (care recipient; yes/no), and type of residence (six options: own house, company housing or official residence, public housing, house of acquaintance (including a family or friend), private rental housing, and other).

Personality factors were taken from two inventories. The "power to live" scale includes eight factors: leadership, problem-solving, altruism, stubbornness, etiquette, emotional regulation, self-transcendence, and active well-being. The questionnaire was comprised of 34 items and each factor had three to five items. The internal consistency and concurrent validity of the questionnaire has been demonstrated [18]. Participants rated the applicability of each description using a 6-point scale (0: not at all; 5: very much). The Big Five personality scale includes extraversion, agreeableness, conscientiousness, neuroticism, and openness, which were measured using the Japanese version of the Ten-Item Personality Inventory [22,23], which includes one positive item and one reverse-scored item for each dimension. We adopted this very short version of the Big Five inventory to minimize respondents' fatigue or frustration, which could decrease the rate and quality of responses. The validity of this short version of the Big Five inventory has been established in terms of convergent and discriminant validity, coverage of sub-dimensions, test-retest reliability, and patterns of external correlates. Participants responded using a 6-point scale (0: not at all; 5: very much) and the scores of reverse items were reverse coded. For each factor or dimension, the sum of the scores was converted to a ratio against the maximum score. Examples of the questions and the statistics for each personality factor or dimension are shown in S2 Table in S1 Results.

## Analysis

The statistical analyses were performed using IBM SPSS Statistics 25 (IBM Corp. Armonk, NY, USA).

**Exploration of major types of difficulty: Cluster analysis.** A cluster analysis was applied to the experience data for the 18 types of difficulty. Experience data on each item of difficulty, which had a range of three possible scores (0: no, 1: a little, or 2: very much), were transformed to binary data by simplifying the score to indicate whether they experienced (1) or did not experience (0) each item. This transformation had the advantage of statistical simplicity, and it also allowed us to avoid potential bias related to subjective criteria for dissociating '1: a little' and '2: very much,' which might have affected the eventual resolution. We applied a hierarchical cluster analysis to the binary-transformed experience data for the 18 items of difficulty (n = 1145, excluding missing data) using Ward's method (which minimizes the total within-cluster variance) and squared Euclidean distance as a dissimilarity measure. The identified cluster was considered as a distinct difficulty type, and the degree of difficulty experienced was defined by the number of items in the cluster. We expected to obtain two or three clusters, with each cluster having five or more items of difficulty (to make it meaningful to use the number of items as an index of the degree).

To characterize each identified difficulty type, the degree of difficulty experienced and the degree of successful resolution using each of the six resolution types was calculated for each participant, and then the distributions were assessed.

**Exacerbating factors for each difficulty type: Multiple regression analysis.** We performed a three-block hierarchical multiple regression analysis [24] on the degree of difficulty experienced (i.e., entered as a dependent variable) separately for each identified difficulty type. First, we explored background factors using a stepwise approach in blocks 1 and 2. Then, we determined whether each personality factor had additional explanatory power in block 3. The statistical threshold for exploratory analyses was set at $p < 0.05$, corrected for multiple

comparisons using the Bonferroni method (*adjusted p*-value = *raw p*-value × number of tests [25]) based on the number of independent variables entered in the analyses. The results of uncorrected $p < 0.05$ were also reported for reference purposes.

In terms of background factors, damage and demographic factors (Table 1) were entered in separate blocks. As damage factors (block 1), eight variables were included. Since damage to home buildings and household goods were strongly correlated (Spearman's ρ = .799), we included only one of them, choosing the latter because it was correlated slightly better with the degrees of all the identified difficulty types. Household goods and injury to oneself were coded as ordinal variables according to the severity of damage. Car was binary coded as 1 for lost or not owning and 0 for an available car. Family injury was binary coded as 1 when any of the family members were injured badly enough to go to the hospital or to be hospitalized, and family death was coded as 1 when any of the family members died. Similarly, injury to a friend was coded as 1 when the injury was bad enough to go to the hospital or to be hospitalized, and the death of a friend was coded as 1 when a friend died. Experiencing refugee life was coded as 1 when the response was yes. As demographic factors (block 2), 23 variables were included. Sex was coded as 1 for male. Age, educational background, and household income were coded as ordinal variables according to the amount or degree. For occupation, household structure, and type of residence, a binary variable was made for each of the options (seven, five, and six, respectively) and coded as 1 when it was selected. The care recipient was coded as 1 when the response was yes. In blocks 1 and 2, the variables were initially selected using a stepwise method (forward selection: $p < 0.05$, backward selection: $p > 0.10$) with limited samples, including the perfect data set for background factors (i.e., the sample was excluded if any data were missing). The analysis was then performed again for the statistical test using all samples that included the selected variables. Correction for multiple comparisons was performed based on the number of independent variables, which was 31.

For the personality factors, the score of each personality factor was entered in block 3. Each personality trait variable was entered separately, that is, 13 separate analyses were performed on block 3 rather than entering the 13 variables altogether. This was to avoid collinearity due to the relatively high correlations across personality trait scores. Correction for multiple comparisons was performed based on the number of independent variables, which was 13.

**Facilitative factors for resolving difficulty: Multiple regression analysis.** We performed a hierarchical multiple regression analysis on the degree of successful resolution (i.e., entered as a dependent variable) separately for each of the six resolution types (i.e., self, request, family, acquaintance, cooperation, and public) for each difficulty type. Similar to the analysis of the exacerbating factors for each difficulty type, we first identified contributing background factors (blocks 1 and 2) and then personality factors (block 3). Since the majority of the participants reported a high degree of experience for one of the identified difficulty types, those with a low degree of experience were excluded from the analysis to take advantage of a higher homogeneity of the samples (see Results for the details). The degree of difficulty experienced was included in the background factor's damage factor (32 variables in total). The policy for the statistical threshold was the same as the analysis of the exacerbating factors for each difficulty type.

## Results

### Major types of difficulty

Fig 1 shows the results of the cluster analysis. The dendrogram (Fig 1A) shows a two-cluster solution with each cluster consisting of nine items. Their plot on a two-dimensional space (Fig 1B) shows the relationships within each cluster and between two clusters. Cluster 1 was

**(a)** Cluster analysis                               **(b)** Multidimensional scaling

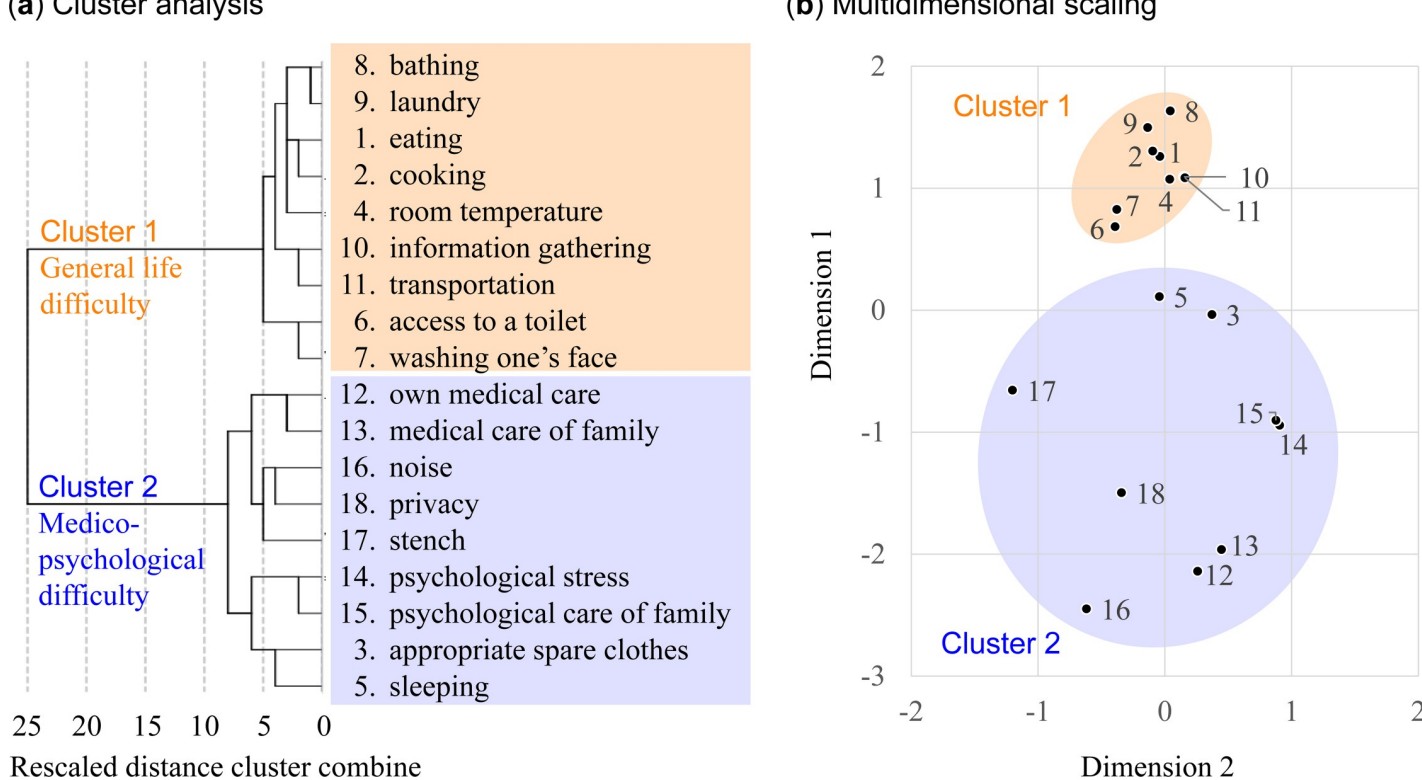

**Fig 1. Results of the cluster analysis.** Dendrogram (**a**) and the distribution of items in a two-dimensional space using multidimensional scaling (**b**) are shown.

composed of the items eating, cooking, and room temperature at its center and included access to a toilet, washing one's face, bathing, laundry, information gathering, and transportation. We labeled this cluster "general life difficulty" because it includes items related to basic life needs that are essential for survival. The items in Cluster 1 were distributed closely to each other, suggesting that similar classes of people experienced most of these troubles, while others experienced them only a little. Cluster 2 had privacy at its center and medical care for oneself and family members, individual and family psychological stress, noise, stench, appropriate spare clothes, and sleeping at the periphery. We labeled this cluster "medico-psychological difficulty" because the items are likely to be related to, or dependent on, individual or household medical or psychological vulnerability. The distribution of items in Cluster 2 was relatively sparse, suggesting that the experience of the items was dependent on various individual factors.

Fig 2 shows the degree of difficulty experienced for each difficulty type and its relationship to successful resolution for each of the six resolution types. General life difficulty was experienced to a high degree (i.e., most of the items were experienced) by most of the participants; 48% experienced all nine items and 85% experience more than half (i.e., five items). On the other hand, the degree of experienced medico-psychological difficulty varied largely across participants. Still, the fact that many participants experienced multiple items (48% experienced more than half the items) may suggest the existence of common exacerbating factors. The correlation of the degrees of difficulty experienced for the two types was modest (Spearman's $\rho$ = .579). The degrees of successful resolution of the six resolution types were different for the two difficulty types. For both difficulty types, self and family solutions contributed around 30% to 40% (Self: 33 ± 32 and 34 ± 36%; Family: 38 ± 34 and 30 ± 34%, respectively for general life

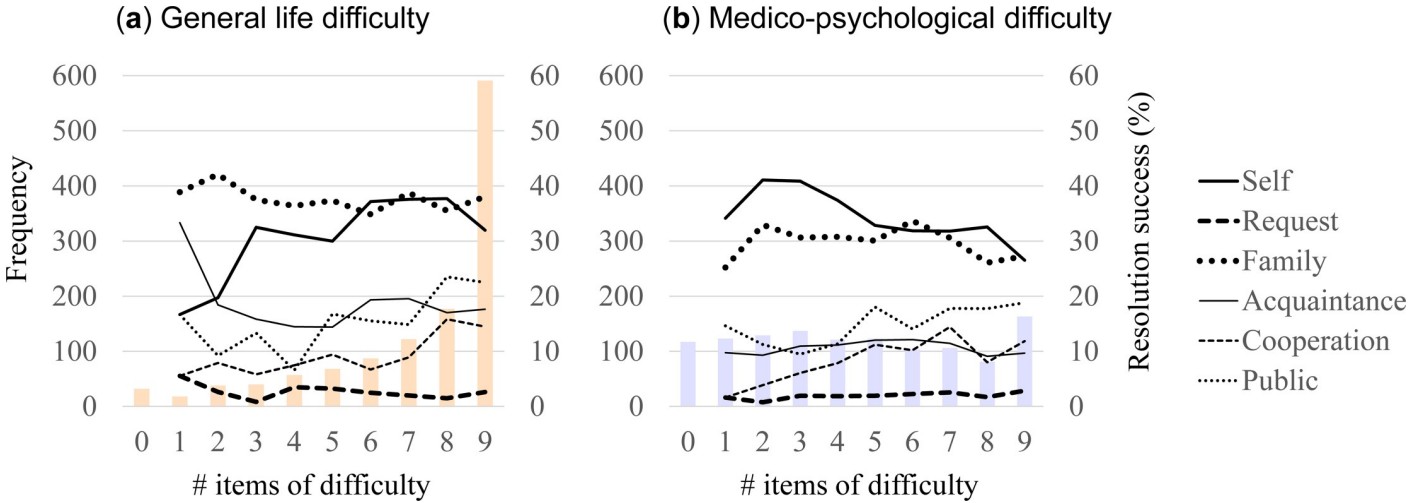

**Fig 2. Characteristics of two difficulty types.** Distribution of the degree (frequency) of difficulty experienced (bar graph) and its relationship to the solution success rate (%), with different lines representing the six solution types (see legend at right), are shown for general life difficulty (**a**) and medico-psychological difficulty (**b**).

difficulty and medico-psychological difficulty); acquaintance, cooperation, and public assistance around 10% to 20% (Acquaintance: 18 ± 25 and 11 ± 21%; Cooperation: 12 ± 21 and 8 ± 19%; Public: 19 ± 25 and 15 ± 24%); and request less than 10% (2 ± 9 and 2 ± 9%). The contribution of cooperation and public assistance appeared to have a tendency to increase as the degree of difficulty increased in both difficulty types (statistically tested later).

Table 2 summarizes the results of regression analyses for exacerbating factors (See S3 and S4 Tables in S1 Results for details). Both difficulty types were affected by several background factors. Damage to household goods was an exacerbating factor for both difficulty types. Damage to car and injury to oneself or one's family exacerbated general life and medico-psychological difficulties, respectively. Living in a single household and being a worker, respectively, were mitigating factors for the two types. Personality factors only affected medico-psychological difficulties; extraversion and neuroticism were mitigating and exacerbating factors, respectively.

## Facilitative factors for rosolving difficulty

We limited our analysis of facilitative factors for resolving general life difficulties to participants (85%) who had experienced more than half (five) of the nine items of difficulty, expecting an advantage in statistical sensitivity from the higher homogeneity of the sample.

**Table 2. Exacerbating factors for two difficulty types.**

| Background factors | | Personality factors | |
|---|---|---|---|
| General life difficulty | Medico-psychological difficulty | General life difficulty | Medico-psychological difficulty |
| Household goods | Household goods | | *Extraversion* |
| Car | | | Neuroticism |
| | Own injury | | |
| | Family injury | | |
| | *Worker* | | |
| *Single* | | | |

Background (damage and demographic) and personality factors that significantly contributed to the degree of difficulty experienced are summarized separately for the two difficulty types. Italicized font denotes a negative contribution (i.e., mitigating factors). See S3 and S4 Tables in S1 Results for further details.

**Table 3. Facilitative factors for six resolution types.**

| | Background factors | | Personality factors | |
|---|---|---|---|---|
| | General life difficulty | Medico-psychological difficulty | General life difficulty | Medico-psychological difficulty |
| Self-resolution (S5 and S6 Tables in S1 Results) | *Household goods* | | Problem solving | |
| | *Car* | | | |
| | *Refugee life* | *Refugee life* | | |
| Resolution through request (S7 and S8 Tables in S1 Results) | | | Altruism | |
| | | | Stubbornness | |
| Resolution through family (S9 and S10 Tables in S1 Results) | *Sex (male)* | | | |
| | *Age* | *Age* | | |
| | Three generation | Three generation | | |
| | Own house | | | |
| Resolution through acquaintance (S11 and S12 Tables in S1 Results) | | *Age* | Leadership | Leadership |
| | Education | | | Extraversion |
| Resolution through cooperation (S13 and S14 Tables in S1 Results) | Refugee life | Refugee life | | |
| | | Difficulty experienced | | |
| Resolution through public assistance (S15 and S16 Tables in S1 Results) | Household goods | | Emotional regulation | |
| | Refugee life | Refugee life | *Neuroticism* | |

Background (damage and demographic) and personality factors that significantly contributed to successful resolution using the six resolution types are summarized separately for the two difficulty types. Italicized font denotes a negative contribution (i.e., a preventative factor). See S5-S16 Tables in S1 Results for further details.

Table 3 summarizes the results of regression analyses for facilitative factors for six resolution types, separately for each difficulty type (See S5-S16 Tables in S1 Results for details). For self-resolution, only a facilitative personality factor was identified for general life difficulty, namely problem solving. Several damage factors prevented the solution: refugee life for both difficulty types and damages to household goods or a car in the case of general life difficulty.

Among the four mutual-assistance resolutions (i.e., resolutions thorough request, family, acquaintance, or cooperation), facilitative factors largely varied. For successful resolution by requesting help, the only facilitative factors identified for general life difficulty were the personality factors altruism and stubbornness. In the case of resolution through one's family, only background factors played a role; being young and living in a three-generation household facilitated the resolution of both difficulty types, and being female and living in one's own house further facilitated the resolution of general life difficulties. Successful resolution through the help of an acquaintance was affected by both background and personality factors; education and leadership facilitated the resolution of general life difficulties, whereas youth, leadership, and extraversion facilitated the resolution of medico-psychological difficulties. Resolution through cooperation was only associated with damage factors; being a refugee was associated with both difficulty types, and a higher degree of experienced difficulty was associated with medico-psychological difficulties.

Resolution through public assistance was associated with both damage and personality factors. Being a refugee was associated with resolution of both difficulty types, and damage to household goods was additionally associated with the resolution of general life difficulty. Emotional regulation facilitated and neuroticism prevented the resolution of general life difficulty.

## Discussion

We explored individual background and personality factors that facilitate self-aid or mutual assistance in the immediate aftermath of a disaster considering different types of difficulty and

resolution. We first identified two types of difficulties: general difficulties in meeting basic life needs, such as food (eating and cooking), body temperature regulation (room temperature), and sanitation (toilet, washing one's face, bathing, and laundry), which were experienced by most victims; and difficulties in meeting medico-psychological needs, such as medical care, psychological stress, privacy, and noise, which varied across victims. Disaster damage was an exacerbating factor for both difficulty types, and medico-psychological difficulties were additionally affected by personality factors, with neuroticism and extraversion acting as exacerbating and mitigating factors, respectively. Then we explored factors that facilitated such resolution. In the case of general life difficulty, self-resolution and resolution through request were facilitated solely by personality factors for problem solving and altruism/stubbornness, respectively. Resolution thorough acquaintances was facilitated by several demographic and personality factors, including education, youth, leadership, and extraversion, whose roles varied between the two difficulty types. Resolution thorough family was facilitated only by demographic factors such as being female or young and living in a three-generation household or one's own home. Disaster damage, particularly the loss of household goods and experiencing refugee life, hampered self-resolution and forced a reliance on cooperation or public assistance.

Our results demonstrated the importance of dissociating difficulty and resolution types when formulating pro-survival individual factors in this disaster stage. Individual exacerbating factors differed among the difficulty types, and facilitative factors also differed across the six resolution types. The existing controversy over the contribution of some individual factors, such as age, sex, and education, may be explained by the fact that, in previous studies, vulnerability factors were explored without such dissociations [10,11]. Our results show that sex, age, and education affected only limited types of resolution success (i.e., family and acquaintance resolutions). The difference in the available type of resolution across disasters, communities, or households, therefore, determines the impact of these individual factors. In fact, the types of available resolutions were affected by some of the damage factors (e.g., loss of household goods and refugee life) and household structure. A constellation of personality factors that influence different types of difficulty resolution was also revealed for the first time by focusing on each resolution type for each type of difficulty.

Our results substantiate the different characterizations of victims presented in the social science and mental health literature: the two broad types of difficulty appear to be largely congruent with the two images. The general life difficulty seems to represent our stereotypical image of the difficulties victims commonly experience in the immediate aftermath of massive disasters, which ubiquitously affected the victims. It therefore makes sense that experiencing this type of difficulty directly affects the summary outcome of the disaster, and social science predominantly examines this type of difficulty to develop vulnerability indices. The observation that single-household victims experienced this type of difficulty to a lesser extent appears reasonable considering that a large household is likely to have children or elderly, who are more vulnerable to disasters [9,11]. On the other hand, the medico-psychological type of difficulty aligns with the manifestations of poor mental health outcomes and related stressors described in the health science literature. Apart from psychological stress and sleep disturbances, difficulties in the areas of privacy, spare clothes, noise, and stench may also be recognized as mental health outcomes in terms of interpersonal or environmental problems in living conditions [13]. These difficulties differ from general life difficulties in that they cause harm through psychological processes. Empirical evidence exists for the association of psychological distress with sensitivity to noise [26] or odor [27]. The other difficulty items, i.e., the need for personal or family medical care, may reflect pre-disaster medical conditions or injuries related to the disaster; both are known as stressors [13]. Importantly, the identification of high neuroticism

and low extraversion as exacerbating these types of difficulties is congruent with the mental health literature [12–14], as is the mitigation of these difficulties by being a worker, given the higher level of social embeddedness and support in that type [13]. Also consistent with both sets of literature is that both types of experienced difficulties were commonly exacerbated by a number of damage factors [10,12–14].

Contrary to the commonly held view that low socioeconomic status is a strong vulnerability factor, income was not identified as a mitigating factor of any difficulty type or as a facilitative factor for any resolution type. Our focus on the immediate aftermath may explain this apparent discrepancy. The conventional view has been established through previous studies, which comprehensively addressed the wide range of post-disaster phases, from immediate aftermath to life recovery [9–11]. Socioeconomic status may play little role in the immediate aftermath, where difficulties are mostly caused by the shortage of life-critical resources due to the disruption of logistical systems, which is not something that can be solved with money. This is in contrast to the long-lasting life-recovery phase, where costly housing reconstruction is a central issue [28,29]. It is also important to note that, compared with previous studies, our victims were relatively well off (i.e., only 12% had yearly income less than 2 million JPY, or approximately 18,000 USD; Table 1); even after the logistical system was restored, most victims had sufficient financial resources for life-critical factors.

This study revealed two important features concerning the contribution of individual psycho-behavioral characteristics to survival in the immediate aftermath of a disaster. First, different personality factors facilitate different types of resolution. The multiplicity of survival-oriented psycho-behavioral characteristics may reflect the complexity of social dynamics as well as the psychological processes involved in survival in a disaster context. It also appears consistent with the emerging socio-ecological view of resilience, which assumes that resilience develops and functions in complex interactions between individuals and their environments [30]. Second, the effect of personality factors on resolution success was mostly limited to general life difficulties. Speculatively, resolution of medico-psychological difficulties requires different processes than does resolution of general life difficulties. Alternatively, medico-psychological difficulties might prevent survival-oriented psycho-behavioral characteristics from functioning.

In the context of disaster risk mitigation focused on evaluating risk and fostering citizen resilience, each personality factor implicated in difficulty-exacerbating or resolution-facilitative approaches is theoretically a target for intervention. A classic approach attempted to include psycho-behavioral factors in explaining different levels of disaster risk between two population of residents in the context of tornado evacuation [31]. Inclusion of the personality factors identified in the current study might contribute to more sophisticated risk evaluation among citizens in the emergency stage. To enhance community resilience, although all the identified personality factors are potential educational targets, we would suggest focusing attention on leadership. Cultivating this trait would promote the ability to resolve difficulties beyond the household or family sphere by more readily asking for help from acquaintances, and it could also help in dealing with medico-psychological difficulties. Strong leadership traits can be valuable as a disaster unfolds; our previous study demonstrated the important role of leadership in self-help and mutual assistance during an impending tsunami evacuation [20,21]. The possible malleability of the leadership personality trait is an important additional consideration. Experience-dependent malleability is suggested by the age-dependent increase of the leadership score [18]. The leadership items describe social behaviors (i.e., gathering people, communication, and maintaining relationships) that may be enhanced by social skill training. Anthropological hypotheses assume the association of leadership with norm-based morality and group-mindedness and its development through interaction with peers [32]. It

seems worth testing whether a daily community activity incorporating such features could enhance the leadership capabilities of community members. If successful, this could be a prime example of people-centered preventive approaches to disaster risk, as encouraged in the Sendai Framework 2015–2030 [8].

This study has certain limitations. The survey was conducted nearly three years after the disaster, which may have affected victims' memories and biases. We also concede two major shortcomings of the survey. First, we could not objectively define a period of time to constitute the "immediate aftermath." It was instead dependent on each victim's individual interpretation of "The restart of a normal life or entering temporary housing." We regret that we failed to ask the actual period of time each responder associated with that definition. Second, the list of difficulty items (i.e., 18 items) and resolution types were, for the most part, arbitrarily determined without a clear basis in the literature. Although we are confident that we covered major items and types based on our experiences as victims and our expertise as disaster researchers, we may have excluded something or failed to choose the most appropriate wording. There were two major limitations in our interpretation of contributing personality factors. First, the list of personality factors that we assessed was obviously not all-inclusive. In particular, self-efficacy and locus of control, which have previously been implicated in disaster survival [33,34] or life recovery [35], should be included in future studies for a more comprehensive picture of the personality correlates of disaster survival. Second, the causal relationship between personality factors and experience of difficulty or its resolution should be carefully examined, as self-evaluation of one's own personality may be biased. It is also known that personality traits may themselves be affected by experiencing a disaster [36,37].

## Conclusions

Analyzing the data from the 2011 Great East Japan Earthquake and Tsunami Disaster, we explored individual background and personality factors that facilitate self-aid or mutual assistance in the immediate aftermath of a disaster considering different types of difficulty and resolution. We first identified two types of difficulties: general life difficulties, which were experienced by most victims, and medico-psychological difficulties, which were experienced to varying degrees across victims. Both types were exacerbated by disaster damages, whereas medico-psychological difficulties were uniquely affected by personality factors, with neuroticism and extraversion acting as exacerbating and mitigating factors, respectively.

Then we examined the effect of individual factors on success in resolving the two types of difficulty through six types of resolution: self-resolution, four types of mutual assistance, and public assistance. In general, disaster damage hampered self-resolution and forced a reliance on cooperation or public assistance. Some demographic factors, such as being young and living in a three-generation household, facilitated resolution thorough the family. Importantly, several personality factors facilitated different types of resolution, primarily of general life difficulties: problem solving facilitated self-resolution, altruism or stubbornness facilitated resolution through request, leadership facilitated resolution through acquaintances, and emotion regulation facilitated resolution through public assistance.

Our results demonstrate the importance of dissociating difficulties and resolution types when identifying facilitative factors for self-aid and mutual assistance in this disaster stage. The identified involvement of different individual factors, particularly personality factors, in different types of resolution may reflect the complexity of social dynamics as well as the psychological processes involved in survival in this context. The current findings contribute to the improvement of disaster risk mitigation through risk evaluation and the fostering of resilience among citizens by allowing separate consideration of different difficulty types and resolution processes.

## Supporting information

**S1 Results. Supplementary results.** Including S1-S16 Tables.
(DOCX)

**S1 Raw data. Raw data set.**
(XLSX)

**S1 File. Original survey (in Japanse; English translation of the relevant part is available in S1 Raw data).**
(PDF)

## Author Contributions

**Conceptualization:** Motoaki Sugiura, Tsuneyuki Abe, Rui Nouchi, Akio Honda, Shosuke Sato, Toshiaki Muramoto, Fumihiko Imamura.

**Data curation:** Akio Honda.

**Formal analysis:** Motoaki Sugiura, Ryo Ishibashi, Tsuneyuki Abe.

**Funding acquisition:** Motoaki Sugiura.

**Investigation:** Motoaki Sugiura, Tsuneyuki Abe, Shosuke Sato.

**Project administration:** Motoaki Sugiura.

**Supervision:** Fumihiko Imamura.

**Writing – original draft:** Motoaki Sugiura.

**Writing – review & editing:** Motoaki Sugiura, Ryo Ishibashi, Tsuneyuki Abe, Rui Nouchi, Akio Honda, Shosuke Sato, Toshiaki Muramoto, Fumihiko Imamura.

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
