## [Decision Letter · Decision Letter 0]

12 Apr 2021

PONE-D-20-29963

Self-help and mutual assistance in the aftermath of a tsunami: how individual factors contribute to resolving difficulties

PLOS ONE

Dear Dr. Sugiura,

Thank you for submitting your manuscript to PLOS ONE. After careful consideration, we feel that it has merit but does not fully meet PLOS ONE’s publication criteria as it currently stands. Therefore, we invite you to submit a revised version of the manuscript that addresses the points raised during the review process.

We look forward to receiving your revised manuscript.

Kind regards,

Dr. Md Nazirul Islam Sarker

Academic Editor

PLOS ONE

Additional Editor Comments:

The author is advised to address all the comments point-by-point and marked in the revised version by track changes or using different color.

Journal Requirements:

2. Please include a copy of Table 12 which you refer to in your text on page 29.

Reviewers' comments:

Reviewer's Responses to Questions

**Comments to the Author**

1. Is the manuscript technically sound, and do the data support the conclusions?

Reviewer #1: Yes

Reviewer #2: Yes

2. Has the statistical analysis been performed appropriately and rigorously? 

Reviewer #1: Yes

Reviewer #2: Yes

3. Have the authors made all data underlying the findings in their manuscript fully available?

Reviewer #1: Yes

Reviewer #2: Yes

4. Is the manuscript presented in an intelligible fashion and written in standard English?

Reviewer #1: Yes

Reviewer #2: Yes

5. Review Comments to the Author

Reviewer #1: I read the paper carefully and I wish to support it for publication after minor revision. This is an original work and depth of research is accurate. The study has significance impact on wide range of researcher in this field of disaster management. The manuscript has been written in Standard English. The research question is well defined. In my eyes, the main comments of the paper are as follows:

1) Introduction part, the author may add few points about disaster management cycle, it will help to understand the overall concept of disaster to readers. The authors are request to add research gap and innovation of the study.

2) The methodology part needs to concise rather details presentation and explanation.

3) Results part may concise

4) Conclusions part need to include key findings of the study.

Overall, the paper should need minor revision.

Reviewer #2: Title: Self-help and mutual assistance in the aftermath of a tsunami: how individual factors contribute to resolving difficulties

Manuscript Number: PONE-D-20-29963

General comments

This study has analysed the Self-help and mutual assistance in the aftermath of a tsunami: how individual factors contribute to resolving difficulties. The subject is interesting. However, the manuscript needs to be reviewed (major review) in order to be accepted.

Abstract

• It is not well structured, even the essential points are not clear. It is relatively full of background text rather than specific point. The author is advised to revise the abstract by following the journal style (academic way).

Introduction

• The introduction is not well-organized, and it has been unnecessarily illustrated with irrelevant text.

• The authors missed up this section's sequence, and also, the rationale of this study is not well placed.

• The authors failed to present the core issue and its importance.

• The author is advised to concentrate on the main issue, significance of this study, avoid unnecessary discussions, and write in a logical order.

Materials and Method

• This section is very weak. It requires revised by following essential scientific steps.

• Table-1, 2 can be sorted in accordance with some key aspects like category, questions, coding, frequency and so on; simultaneously the description should be scientific and coherent.

Results and discussion

• Discussion- this part is more descriptive than the explanation of the results. The author should take appropriate and well-organized measures to clarify the study's findings.

Conclusion

• It should be revised and followed the updated version

References

You should critically look at your references like capital letter, small letter, issue, etc.

Decision

Major Revision

6. PLOS authors have the option to publish the peer review history of their article (what does this mean?). If published, this will include your full peer review and any attached files.

Reviewer #1: **Yes: **Md. Nuralam Hossain

Reviewer #2: No

---

## [Author Response · Author response to Decision Letter 0]

5 Jun 2021

Response to Reviewer #1:

C1.1) Abstract and title

C1.1.1) The abstract need to be concise 

R1.1.1) To make the abstract concise, we reduced and reorganized the report of the results as follows (L26–35): 

“We identified general life difficulties and medico-psychological difficulties as two broad types of difficulty; disaster damage contributed to both types, while some personality factors (e.g., neuroticism) exacerbated the latter. Disaster damage hampered self-resolution and forced a reliance on resolution through cooperation or public assistance. On the other hand, some demographic factors, such as being young and living in a three-generation household, facilitated resolution thorough the family. Several personality factors facilitated different types of resolution, primarily of general life difficulties; the problem-solving factor facilitated self-resolution, altruism, or stubbornness resolutions through requests, leadership resolution through acquaintance, and emotion-regulation resolution through public assistance.”

C1.1.2) The study aim is not clear

R1.1.2) We clarified our study aim as follows (L19–21):

“To identify such individual factors in the background (i.e., disaster damage and demographic) and personality domains considering different types of difficulty and resolution, we analyzed…”

C1.1.3) Data collection time acquisition needs to mention

R1.1.3) The time of data collection is now mentioned as follows (L21–22):

“we analyzed survey data collected in the 3-year aftermath of the 2011 Great East Japan Earthquake and Tsunami.”

C1.1.4) The authors are request to add implication of the study

R1.1.4) The implications of the study are now clarified as follows (L35–39):

“Our findings are the first to demonstrate the involvement of different individual, particularly personality, factors in survival in the complex social dynamics of this disaster stage. They may contribute to disaster risk mitigation, allowing sophisticated risk evaluation and community resilience building.”

C1.2) Introduction

C1.2.0) the author may add few points about disaster management cycle, it will help to understand the overall concept of disaster to readers. The authors are request to add research gap and innovation of the study.

R1.2.0) We thank the reviewer for this advice. We included the disaster management cycle perspective and described the significance of our study in this context as follows: 

(L47–50)

“Among the stages of the disaster management cycle (i.e., mitigation, preparation, emergency, and recovery stages), the emergency stage is a critical period for intervention by organizations [4], but such public support takes time and is often limited.”

(L52-54)

“Understanding the individual factors that contribute to successfully resolving difficulties encountered is important for risk evaluation and for fostering citizens’ resilience during times of quiescence, namely, the mitigation stage [4].”

We also clarified gaps in the research in the second and third paragraphs and innovation in the fourth paragraph as follows:

Gap 1: paragraph 2 (L57–58)

“Published research on such individual factors has primarily addressed factors that contribute to negative consequences, that is, exacerbating factors.”

Gap 2: paragraph 3 (L73-86)

“To explore the individual factors that contribute to the resolution of difficulties, the following three issues should be addressed. First, different categories of difficulty may be dealt with separately given the apparently distinct characteristics of victims as represented by the social science and mental health fields... Second, different types of resolution should be addressed separately, as individual factors likely affect the availability of different resolution types… Finally, in addition to common background factors, we were interested in personality traits as individual factors...”

Innovation: paragraph 4 (L87-89)

“In this study, we investigated the individual background and personality factors that facilitate self-aid or mutual assistance in the immediate aftermath of a disaster considering different types of difficulty and resolution.”

C1.2.1) Need to define vulnerability and aftermath/after disaster what kind of vulnerability has to face of people. 

R1.2.1) We recognize that the term “vulnerability” was used inappropriately in the original version of our manuscript to imply any factors that contribute to negative consequences (e.g., disaster damage). In the revised version, we limited the use of this term to factors that are not typically considered direct causes of the consequence, a distinction made elsewhere in the literature (e.g., 9–11). We now use the term “exacerbating factor” when referring to all factors that contribute to negative consequences.

C1.2.2) What is dependent variable?

R1.2.2) For the multiple regression analyses, the degree of difficulty experienced was the dependent variable used to identify exacerbating factors for each difficulty type, and the degree of successful resolution by each of the six resolution types was the dependent variable used to identify the factors facilitating the resolution of a difficulty. These are now clarified as follows:

(L212–214)

“We performed a three-block hierarchical multiple regression analysis [24] on the degree of difficulty experienced (i.e., entered as a dependent variable) separately for each identified difficulty type.”

(L252-255)

“We performed a hierarchical multiple regression analysis on the degree of successful resolution (i.e., entered as a dependent variable) separately for each of the six resolution types (i.e., self, request, family, acquaintance, cooperation, and public) for each difficulty type.”

C1.3) Methods

C1.3.0) The methodology part needs to concise rather details presentation and explanation.

R1.3.0) The Materials and Methods section was made more concise by moving Tables 2 and 3 in the original manuscript to the supplementary section (S1 and S2 Tables in S1 Results) and removing the explanation of the general structure of the analysis, which was already presented in the Introduction. 

C1.3.1) Table 1. Damage and demographic profiles of the participants.; need to concise

R1.3.1) Table 1 was totally reorganized (L130). We removed the actual text of the questionnaire (the original survey is available in the supplementary materials) and put the variable labels, options, and statistics in different columns.

C1.3.2) Table 2. Experiences of difficulty and 166 how they were resolved. Can be put in supplementary part.

R1.3.2) We moved the original Table 2 to the supplementary section (S1 Table in S1 Results).

C1.3.3) Table 3. Personality factors. Need to put in supplementary section.

R1.3.3) We moved the original Table 3 to the supplementary section (S2 Table in S1 Results).

C1.3.4) the authors used a scale to evaluate the degree to participants had experienced difficulty; scale 0= none, 1= Little and 2= very high. Is this scale used previous any studies? If yes, can you please add citation?

R1.3.4) The scale was adopted without reference to any previous literature, not an uncommon practice in surveys. Here, a three-point scale, rather than a binary scale, was employed to encourage responders with a moderate degree of difficulty to select 1, as they would likely select “no” on a binary scale considering other seriously damaged victims. This motivation for adopting the three-point scale is now included as follows (L143–146):

“This was meant to encourage responders with a moderate degree of difficulty to select 1, as they would likely select “none” on a binary scale to differentiate their losses from those of seriously damaged victims.”

C1.3.5) Is there any basis to divided into clusters?

R1.3.5) In the Introduction, we have clarified the background and motivation for using cluster analysis to identify major types of difficulty: 

Background (L74–78)

“First, different categories of difficulty may be dealt with separately given the apparently distinct characteristics of victims as represented by the social science and mental health fields. We were curious as to whether the distinction reflects a lack of integrated research between the two fields [15] or whether there are indeed different categories of difficulty.”

Motivation (L99–100)

“First, we explored the major types of difficulty among the 18 areas using a cluster analysis”

C1.3.6) Line no.-249; need ref.?

R1.3.6) We added a reference for the Bonferroni correction, along with an explanation, as follows (L216–219):

“The statistical threshold for exploratory analyses was set at p < 0.05, corrected for multiple comparisons using the Bonferroni method (adjusted p-value = raw p-value × number of tests [25]) based on the number of independent variables entered in the analyses.”

25. Armstrong RA. When to use the Bonferroni correction. Ophthalmic Physiol Opt. 2014;34(5):502-8. doi: 10.1111/opo.12131.

C1.4) Results

C1.4.0) Results part may concise

R1.4.0) The Results section was made more concise by focusing the discussion of the hierarchical multiple regressions results on the summary tables (now Tables 2 and 3) and moving the original Tables 5–18 to the supplementary section (S3–16 Tables in S1 Results).

C1.4.1) Line no.-308; in Cluster-2; what are the privacy item? Can you add these?

R1.4.1) We apologize for the misleading wording. We were referring to the 18th type of experienced difficulty, as described in the Variables section of the Methods. For clarity, we updated the description as follows (L274-276):

“Cluster 2 had privacy at its center and medical care for oneself and family members, individual and family psychological stress, noise, stench, appropriate spare clothes, and sleeping at the periphery.”

C1.4.2) Line no.-370; need Ref.?

R1.4.3) As a reference for a hierarchical multiple regression analysis including the stepwise selection of variables, we have cited a textbook (R212–216):

“We performed a three-block hierarchical multiple regression analysis [24] on the degree of difficulty experienced (i.e., entered as a dependent variable) separately for each identified difficulty type. First, we explored background factors using a stepwise approach in blocks 1 and 2. Then, we determined whether each personality factor had additional explanatory power in block 3.” 

24. Tabachnick BG, Fidell LS. Multiple Regression. Using Multivariate Statistics. 7th ed. Boston: Pearson Education,Inc.; 2007. p. 99-166.

C1.5) Discussion and Conclusions

C1.5.0) Conclusions part need to include key findings of the study.

R1.5.0) We summarized our key findings in the Conclusion as follows (L491–506).

“…We first identified two types of difficulties: general life difficulties, which were experienced by most victims, and medico-psychological difficulties, which were experienced to varying degrees across victims. Both types were exacerbated by disaster damages, whereas medico-psychological difficulties were uniquely affected by personality factors, with neuroticism and extraversion acting as exacerbating and mitigating factors, respectively. 

Then we examined the effect of individual factors on success in resolving the two types of difficulty through six types of resolution: self-resolution, four types of mutual assistance, and public assistance. In general, disaster damage hampered self-resolution and forced a reliance on cooperation or public assistance. Some demographic factors, such as being young and living in a three-generation household, facilitated resolution thorough the family. Importantly, several personality factors facilitated different types of resolution, primarily of general life difficulties: problem solving facilitated self-resolution, altruism or stubbornness facilitated resolution through request, leadership facilitated resolution through acquaintances, and emotion regulation facilitated resolution through public assistance.”

C1.5.1) What is relationship difficulty with “body temperature regulation” is there any citation?

R1.5.1) This is an interpretation of our results, specifically, the item “room temperature” in cluster 1. This is now clarified by including the relevant items in the discussion of results (L356–359):

“general difficulties in meeting basic life needs, such as food (eating and cooking), body temperature regulation (room temperature), and sanitation (toilet, washing one’s face, bathing, and laundry),”

C1.5.2) Line no-511, the income level didn’t affect the success in any resolution; it is not clear, because pro-survival and post disaster, the individual and country level there is an emergency, so how income level didn’t affect the resolution.

R1.5.2) The argument was based on our results showing that income was not identified as a mitigating factor for any difficulty type or a facilitative factor for any resolution type. This is now clarified as follows (L415–417):

“Contrary to the commonly held view that low socioeconomic status is a strong vulnerability factor, income was not identified as a mitigating factor of any difficulty type or as a facilitative factor for any resolution type.”

We discussed this because, as the reviewer pointed out, the finding is surprising. Our speculation regarding this negative finding is as follows (L421–423):

“Socioeconomic status may play little role in the immediate aftermath, where difficulties are mostly caused by the shortage of life-critical resources due to the disruption of logistical systems, which is not something that can be solved with money.”

C1.5.3) Line no.-540, where is vulnerability indices?

R1.5.3) The description no longer exists, as the paragraph has been removed.

C1.5.4) Line no-570; need citation.

R1.5.4) The argument was based on our own data (Table 1). It is now referenced as follows (L426–427):

“our victims were relatively well off (i.e., only 12% had yearly income less than 2 million JPY, or approximately 18,000 USD; Table 1)”

C1.5.5) the authors are advice to avoid unnecessary discussion because it creates lengthy discussion.

R1.5.5) We removed two paragraphs from the detailed interpretation of the contribution of each personality factor and a summary paragraph from the discussion.

C1.5.6) The authors are advice to discuss the main issue-based analysis what they found. 

R1.5.6) After excluding a lengthy discussion of the personality factors (cf. R1.5.5), we believe our discussion is now focused on the findings. To call attention to the implications of our findings instead, we added the following discussion (L442–448):

“In the context of disaster risk mitigation focused on evaluating risk and fostering citizen resilience, each personality factor implicated in difficulty-exacerbating or resolution-facilitative approaches is theoretically a target for intervention. A classic approach attempted to include psycho-behavioral factors in explaining different levels of disaster risk between two population of residents in the context of tornado evacuation [31]. Inclusion of the personality factors identified in the current study might contribute to more sophisticated risk evaluation among citizens in the emergency stage.”

C1.5.7) Conclusion need to be concise, need to focus on key findings based on analysis

C1.5.7) We improved the conciseness of the Conclusions section by focusing on the key findings (cf. R1.5.0) and clarifying their implications as follows (L507–514): 

“Our results demonstrate the importance of dissociating difficulties and resolution types when identifying facilitative factors for self-aid and mutual assistance in this disaster stage. The identified involvement of different individual factors, particularly personality factors, in different types of resolution may reflect the complexity of social dynamics as well as the psychological processes involved in survival in this context. The current findings contribute to the improvement of disaster risk mitigation through risk evaluation and the fostering of resilience among citizens by allowing separate consideration of different difficulty types and resolution processes.”

C1.6) References

C1.6.1) Few references are not appropriately listed

R1.6.1) We have carefully checked the reference list to assure that it conforms to the journal’s requirements.

C1.6.2) Many references are not recent.

R1.6.2) We checked the literature and added the following newer references in support of two of our arguments. 

- Disaster mental-health literature on contributing individual factors:

14. Goldmann E, Galea S. Mental health consequences of disasters. Annu Rev Public Health. 2014;35(1):169-83. doi: 10.1146/annurev-publhealth-032013-182435.

37. Zoellner T, Maercker A. Posttraumatic growth in clinical psychology - a critical review and introduction of a two component model. Clin Psychol Rev. 2006;26(5):626-53. doi: 10.1016/j.cpr.2006.01.008.

Although several references for other issues are not recent, we believe that the more recent literature offers no fundamental updates to the earlier arguments. 

Response to Reviewer #2:

C2.1) Abstract

It is not well structured, even the essential points are not clear. It is relatively full of background text rather than specific point. The author is advised to revise the abstract by following the journal style (academic way).

R2.1) We totally revised the abstract to clarify the background, purpose, and implications of the study, as follows (L17–39):

“Self-aid and mutual assistance among victims are critical for resolving difficulties in the immediate aftermath of a disaster, but individual facilitative factors for such resolution processes are poorly understood. To identify such individual factors in the background (i.e., disaster damage and demographic) and personality domains considering different types of difficulty and resolution, we analyzed survey data collected in the 3-year aftermath of the 2011 Great East Japan Earthquake and Tsunami. We first identified major types of difficulty using a cluster analysis of 18 difficulty domains and then explored individual factors that facilitated six types of resolution (self-help, request for help, help from family, help from an acquaintance, help through cooperation, and public assistance) of these difficulty types. We identified general life difficulties and medico-psychological difficulties as two broad types of difficulty; disaster damage contributed to both types, while some personality factors (e.g., neuroticism) exacerbated the latter. Disaster damage hampered self-resolution and forced a reliance on resolution through cooperation or public assistance. On the other hand, some demographic factors, such as being young and living in a three-generation household, facilitated resolution thorough the family. Several personality factors facilitated different types of resolution, primarily of general life difficulties; the problem-solving factor facilitated self-resolution, altruism, or stubbornness resolutions through requests, leadership resolution through acquaintance, and emotion-regulation resolution through public assistance. Our findings are the first to demonstrate the involvement of different individual, particularly personality, factors in survival in the complex social dynamics of this disaster stage. They may contribute to disaster risk mitigation, allowing sophisticated risk evaluation and community resilience building.”

C2.2) Introduction

The introduction is not well-organized, and it has been unnecessarily illustrated with irrelevant text. The authors missed up this section's sequence, and also, the rationale of this study is not well placed. The authors failed to present the core issue and its importance. The author is advised to concentrate on the main issue, significance of this study, avoid unnecessary discussions, and write in a logical order.

R2.2) We totally reorganized the Introduction in the standard logical structure of a scientific paper. We particularly emphasized the clarification of research gaps and innovations:

Gap 1: paragraph 2 (L57–58)

“Published research on such individual factors has primarily addressed factors that contribute to negative consequences, that is, exacerbating factors.”

Gap 2: paragraph 3 (L73-86)

“To explore the individual factors that contribute to the resolution of difficulties, the following three issues should be addressed. First, different categories of difficulty may be dealt with separately given the apparently distinct characteristics of victims as represented by the social science and mental health fields... Second, different types of resolution should be addressed separately, as individual factors likely affect the availability of different resolution types… Finally, in addition to common background factors, we were interested in personality traits as individual factors...”

Innovation: paragraph 4 (L87-89)

“In this study, we investigated the individual background and personality factors that facilitate self-aid or mutual assistance in the immediate aftermath of a disaster considering different types of difficulty and resolution.”

C2.3) Materials and Method

This section is very weak. It requires revised by following essential scientific steps.

Table-1, 2 can be sorted in accordance with some key aspects like category, questions, coding, frequency and so on; simultaneously the description should be scientific and coherent.

R2.3) The Materials and Methods section was revised to focus on the essential logical steps; we moved the original Tables 2 and 3 to the supplementary section (S1 and S2 Tables in S1 Results). We also removed our explanation of the general structure of the analysis, which was already noted in the Introduction. Table 1 was totally reorganized (L130). We removed the actual text of the questionnaire (which is available through the original survey in the supplementary section) and put the variable labels, options, and statistics in different columns.

C2.4) Results and discussion

C2.4.1) Discussion- this part is more descriptive than the explanation of the results. The author should take appropriate and well-organized measures to clarify the study's findings.

R2.4.1) The summary of results in the initial paragraph was made more concise; we did not completely remove it in accordance with the suggestion of Reviewer #1. We believe that the rest of the discussion goes beyond mere explanation and addresses the significance or implications of the findings. We also excluded the lengthy discussion of personality factors. We believe that our discussion is now focused on our findings. 

C2.4.2) Conclusion

It should be revised and followed the updated version

R2.4.2) The Conclusion was totally rewritten based on the updated Discussion (L488–514):

“Analyzing the data from the 2011 Great East Japan Earthquake and Tsunami Disaster, we explored individual background and personality factors that facilitate self-aid or mutual assistance in the immediate aftermath of a disaster considering different types of difficulty and resolution. We first identified two types of difficulties: general life difficulties, which were experienced by most victims, and medico-psychological difficulties, which were experienced to varying degrees across victims. Both types were exacerbated by disaster damages, whereas medico-psychological difficulties were uniquely affected by personality factors, with neuroticism and extraversion acting as exacerbating and mitigating factors, respectively. 

Then we examined the effect of individual factors on success in resolving the two types of difficulty through six types of resolution: self-resolution, four types of mutual assistance, and public assistance. In general, disaster damage hampered self-resolution and forced a reliance on cooperation or public assistance. Some demographic factors, such as being young and living in a three-generation household, facilitated resolution thorough the family. Importantly, several personality factors facilitated different types of resolution, primarily of general life difficulties: problem solving facilitated self-resolution, altruism or stubbornness facilitated resolution through request, leadership facilitated resolution through acquaintances, and emotion regulation facilitated resolution through public assistance. 

Our results demonstrate the importance of dissociating difficulties and resolution types when identifying facilitative factors for self-aid and mutual assistance in this disaster stage. The identified involvement of different individual factors, particularly personality factors, in different types of resolution may reflect the complexity of social dynamics as well as the psychological processes involved in survival in this context. The current findings contribute to the improvement of disaster risk mitigation through risk evaluation and the fostering of resilience among citizens by allowing separate consideration of different difficulty types and resolution processes.”

C2.5) References

You should critically look at your references like capital letter, small letter, issue, etc.

R2.5) We have carefully checked the reference list to assure that it conforms to the journal’s requirements.

Response to Journal Requirements

C3.1) Please ensure that your manuscript meets PLOS ONE's style requirements, including those for file naming. The PLOS ONE style templates can be found at

R3.1) We made sure that our revised manuscript meets these requirements before submission.

C3.2) Please include a copy of Table 12 which you refer to in your text on page 29.

R3.2) We apologize for mislabeling this table. It is now included in the supplementary materials (S1 Results) with the label “S10 Table.”

---

## [Decision Letter · Decision Letter 1]

9 Aug 2021

PONE-D-20-29963R1

Self-help and mutual assistance in the aftermath of a tsunami: how individual factors contribute to resolving difficulties

PLOS ONE

Dear Dr. Sugiura,

Thank you for submitting your manuscript to PLOS ONE. After careful consideration, we feel that it has merit but does not fully meet PLOS ONE’s publication criteria as it currently stands. Therefore, we invite you to submit a revised version of the manuscript that addresses the points raised during the review process.

We look forward to receiving your revised manuscript.

Kind regards,

Md Nazirul Islam Sarker

Academic Editor

PLOS ONE

Additional Editor Comments (if provided):

The literature review, method and discussion sections are weak. The author is advised to address all comments point-by-point and improve these sections.

Reviewers' comments:

Reviewer's Responses to Questions

**Comments to the Author**

1. If the authors have adequately addressed your comments raised in a previous round of review and you feel that this manuscript is now acceptable for publication, you may indicate that here to bypass the “Comments to the Author” section, enter your conflict of interest statement in the “Confidential to Editor” section, and submit your "Accept" recommendation.

Reviewer #1: All comments have been addressed

Reviewer #2: All comments have been addressed

Reviewer #3: (No Response)

2. Is the manuscript technically sound, and do the data support the conclusions?

Reviewer #1: Yes

Reviewer #2: Yes

Reviewer #3: Partly

3. Has the statistical analysis been performed appropriately and rigorously? 

Reviewer #1: Yes

Reviewer #2: Yes

Reviewer #3: Yes

4. Have the authors made all data underlying the findings in their manuscript fully available?

Reviewer #1: Yes

Reviewer #2: Yes

Reviewer #3: (No Response)

5. Is the manuscript presented in an intelligible fashion and written in standard English?

Reviewer #1: Yes

Reviewer #2: Yes

Reviewer #3: Yes

6. Review Comments to the Author

Reviewer #1: Section Re-Review comments

Abstract, title

and references -The authors have reduced, reorganized and concise the abstract properly.

-they clarified the study aim in the revised version.

-the authors have mentioned the data collection time acquisition time.

-The authors have rewritten the implication of the research findings properly

Introduction/ Background

-The authors have added some text about disaster management cycle and written research gap with innovation part properly.

-the author properly clarified the issues of aftermath/after disaster what kind of

vulnerability has to face of people.

-they used multiple regression analysis.

Methods

-The authors have revised the The Materials and Methods section and concise by moving Tables 2 and 3 in the original manuscript to the supplementary section

-they addressed other issues raised by reviewer

Results

-The authors have concise this part according to the reviewer suggestion.

-they have addressed other issues marked by reviewer

Discussion and Conclusions

-the author has revised and concise the section based on reviewer comments.

-the authors read carefully all other comments and revised the manuscript accordingly.

-They have updated the reference section and followed the Journal guidelines.

Overall

I’ve read the author responses/revision and the revised manuscript whether they have addressed the issues properly or not. Based on my understanding the authors have addressed all the comments and revised their manuscript appropriately.

Overall, the paper may consider for publication after minor revision.

Reviewer #2: I looked at the manuscript carefully and I found that the author responded appropriately to each of the raised questions. I hope that it is now suitable to be published.

Reviewer #3: The study examines the relationship between individuals’ contributions to the tsunami effects. While the paper offers an interesting understanding, it needs more work to be published.

Specific comments:

The abstract needs a smooth transition through the tsunami and individual aspects. In the current edition, there is an abrupt jump to the study without a clear concentration.

Abstract: It has a lot of irrelevant information that should not be here. Just mention the study aim and key findings, not the whole story and every single finding or discussion here.

Introduction: What is self-aid and mutual assistance? The reader needs a great explanation.

What are the individuals factors the author(s) believe studied poor? Also, it is so early to start what is not “studied well” in this section. Mention what the key missing parts by providing some “studied” aspects.

p.5: 18 areas or 18 categories?

Can you please refine the core study aims of the research as it seems there is no clear concentration? Please, highlight the study contributions clearly.

I have not seen any literature review section!? I am pretty sure there are some studies to be covered.

Method: Is the data collected from 2011 year or 3600 people who experienced 2011? If the answer is former, why did this study conducted so late? If answer is the latter one, when was this survey conducted?

How did the author(s) collect the survey questions? By email, by post service, or some 3rd party organization or data source that the author(s) use it for this study?

Why did the author(s) select 18 categories? Why not 15 or 20?

What is the DV in this study? You observed the 18 categories on what?

Analysis: How come the study was performed with hierarchical regression? What are the motives for this? The reader should be prepared for this in earlier section(s).

Some of the tables should be condensed or removed or reorganized. It seems too fluffy by having more than 15 tables! More like a report I would say rather than a scientific paper.

Same comment applies for the discussion. Just discuss the main findings. It seems too long.

On the other hand, conclusion should be highlighted as it is the most important part of this study.

7. PLOS authors have the option to publish the peer review history of their article (what does this mean?). If published, this will include your full peer review and any attached files.

Reviewer #1: **Yes: **Md. Nuralam Hossain

Reviewer #2: No

Reviewer #3: No

---

## [Author Response · Author response to Decision Letter 1]

7 Sep 2021

Response to Reviewer #1:

C1.1) Abstract, title and references 

-The authors have reduced, reorganized and concise the abstract properly.

-they clarified the study aim in the revised version.

-the authors have mentioned the data collection time acquisition time.

-The authors have rewritten the implication of the research findings properly

C1.2) Introduction/ Background

-The authors have added some text about disaster management cycle and written research gap with innovation part properly.

-the author properly clarified the issues of aftermath/after disaster what kind of vulnerability has to face of people.

-they used multiple regression analysis.

C1.3) Methods

-The authors have revised the The Materials and Methods section and concise by moving Tables 2 and 3 in the original manuscript to the supplementary section

-they addressed other issues raised by reviewer

C1.4) Results

-The authors have concise this part according to the reviewer suggestion.

-they have addressed other issues marked by reviewer

C1.5) Discussion and Conclusions

-the author has revised and concise the section based on reviewer comments.

-the authors read carefully all other comments and revised the manuscript accordingly.

-They have updated the reference section and followed the Journal guidelines.

C1.6) Overall

I’ve read the author responses/revision and the revised manuscript whether they have addressed the issues properly or not. Based on my understanding the authors have addressed all the comments and revised their manuscript appropriately.

C1.7) Overall, the paper may consider for publication after minor revision. 

R1) We are happy that we could respond appropriately to all the comments from this reviewer. We thank them again for their constructive comments. While C1.7 may be read as there is something left for minor revision, we understand that it is meant for the points raised by other reviewers since no specific issues are raised by this reviewer.

Response to Reviewer #2:

C2) I looked at the manuscript carefully and I found that the author responded appropriately to each of the raised questions. I hope that it is now suitable to be published..

R2) We are very pleased to be able to respond appropriately to all the comments from this reviewer. We thank them again for their constructive comments.

Response to Reviewer #3:

C3.0) The study examines the relationship between individuals’ contributions to the tsunami effects. While the paper offers an interesting understanding, it needs more work to be published.

R3.0) We thank the reviewer for their constructive suggestions. We suppose these are based on the first version of our manuscript (ver. 1). We believe that we have already addressed many of the issues raised in this review in the first extensive revision (ver.2), which addressed a number of comments from the two reviewers. However, some issues were newly pointed out, and we have made new responses to them in this revised version (ver.3). In the revised version (ver. 3) including the revision history, the first revision corresponding to the issues pointed out this time is indicated in blue letters, and the new revision this time is indicated in red letters.

C3.1) Abstract:

C3.1.1) The abstract needs a smooth transition through the tsunami and individual aspects. In the current edition, there is an abrupt jump to the study without a clear concentration. 

R3.1.1) In the first sentence of the abstract, by focusing on the core issues only, we made a smooth transition from the academic background to our research motivation as follows (L17–19):

“Self-aid and mutual assistance among victims are critical for resolving difficulties in the immediate aftermath of a disaster, but individual facilitative factors for such resolution processes are poorly understood.”

C3.1.2) It has a lot of irrelevant information that should not be here. Just mention the study aim and key findings, not the whole story and every single finding or discussion here.

R3.1.2) We now clarified the research purpose and removed some descriptions that might have disturbed readers’ understanding. While we could not diminish the results given the exploratory nature of the study and the study’s significance in terms of the complexity of the results, we believe that such an intention is now clear in the revised abstract as follows (L19–37): 

“To identify such individual factors in the background (i.e., disaster damage and demographic) and personality domains considering different types of difficulty and resolution, we analyzed survey data collected in the 3-year aftermath of the 2011 Great East Japan Earthquake and Tsunami. We first identified major types of difficulty using a cluster analysis of 18 difficulty domains and then explored individual factors that facilitated six types of resolution (self-help, request for help, help from family, help from an acquaintance, help through cooperation, and public assistance) of these difficulty types. We identified general life difficulties and medico-psychological difficulties as two broad types of difficulty; disaster damage contributed to both types, while some personality factors (e.g., neuroticism) exacerbated the latter. Disaster damage hampered self-resolution and forced a reliance on resolution through cooperation or public assistance. On the other hand, some demographic factors, such as being young and living in a three-generation household, facilitated resolution thorough the family. Several personality factors facilitated different types of resolution, primarily of general life difficulties; the problem-solving factor facilitated self-resolution, altruism, or stubbornness resolutions through requests, leadership resolution through acquaintance, and emotion-regulation resolution through public assistance. Our findings are the first to demonstrate the involvement of different individual, particularly personality, factors in survival in the complex social dynamics of this disaster stage.”

C3.2) Introduction:

C3.2.1) What is self-aid and mutual assistance? The reader needs a great explanation.

R3.2.1) We clarified the context of the self-aid and mutual assistance addressed in this study by additionally explaining their roles in the disaster management cycle as follows (L45–51):

“Victims of natural disasters experience a variety of difficulties in the immediate aftermath, such as access to food, water, sanitation, and medical assistance, as well as psychological problems [1-3]. Among the stages of the disaster management cycle (i.e., mitigation, preparation, emergency, and recovery stages), the emergency stage is a critical period for intervention by organizations [4], but such public support takes time and is often limited. Self-aid and mutual assistance among victims in resolving such difficulties are critical for survival before public support becomes available [5-7].”

C3.2.2) What are the individuals factors the author(s) believe studied poor? Also, it is so early to start what is not “studied well” in this section. Mention what the key missing parts by providing some “studied” aspects.

R3.2.2) We reorganized the relevant paragraphs. The new second paragraph is now dedicated to the review of what has previously been studied as follows (L57–72):

“Published research on such individual factors has primarily addressed factors that contribute to negative consequences, that is, exacerbating factors. In this regard, the social science and mental health literatures have provided rather different images of victims. In the social science literature, the image of victims is rather vague in terms of the difficulties they encounter. Proposals of vulnerability factors [9-11] have been based on reviews of studies that describe vulnerability in association with various outcomes throughout the disaster stages, including casualty, physical or mental health, post-disaster socioeconomic states, and disaster preparedness. They commonly point out low socioeconomic status as a significant vulnerability factor. On the other hand, the image of the victim is clear in the mental health literature, primarily addressing poor mental health outcomes, such as post-traumatic stress disorders (PTSD). Although various individual factors have also been identified as risk factors, robust examination of disaster exposure and psychosocial resources (e.g., psychiatric pre-morbidity and personality) seems to be unique to the mental health literature [12-14]. In both literatures, the contributions of several common individual factors such as age, sex, and education are controversial.”

 Then we sorted out the poorly studied aspects of the individual factors in the third paragraph as follows (L73–86):

“To explore the individual factors that contribute to the resolution of difficulties, the following three issues should be addressed. First, different types of difficulty may be dealt with separately given the apparently distinct characteristics of victims as represented by the social science and mental health fields. We were curious as to whether the distinction reflects a lack of integrated research between the two fields [15] or whether there are indeed different types of difficulty. Second, different types of resolution should be addressed separately, as individual factors likely affect the availability of different resolution types. For example, self-help may be limited by disaster damage, mutual assistance between familiar people may depend on existing social capital, and that between unfamiliar people may require an explicit request or proposal [16, 17]. Finally, in addition to common background factors, we were interested in personality traits as individual factors. Understanding personality traits, or individual psycho–behavioral characteristics, may be a promising approach for educational or cultural investigations into improving resilience [7, 11].”

C3.2.3) p.5: 18 areas or 18 categories?

R3.2.3) We meant 18 ‘areas’ when we address different items of difficulty in the original survey. The term ‘category’ in the original manuscript was meant for a group of such difficulty items that share some attribute; we used ‘types’ for this term in the revised manuscript. We hope that the meanings of these two terms are now clear in the revised manuscript. 

C3.2.4) Can you please refine the core study aims of the research as it seems there is no clear concentration? Please, highlight the study contributions clearly.

R3.2.4) We clarified the core study aims of the research in the fourth paragraph of the revised Introduction as follows (L87–89):

“In this study, we investigated the individual background and personality factors that facilitate self-aid or mutual assistance in the immediate aftermath of a disaster considering different types of difficulty and resolution.”

 This part is preceded by a paragraph where our research questions are sorted into three issues accompanied by the significance of their clarification as follows (L73–86; c.f., R3.2.2):

“To explore the individual factors that contribute to the resolution of difficulties, the following three issues should be addressed. First, different types of difficulty may be dealt with separately given the apparently distinct characteristics of victims as represented by the social science and mental health fields. We were curious as to whether the distinction reflects a lack of integrated research between the two fields [15] or whether there are indeed different types of difficulty. Second, different types of resolution should be addressed separately, as individual factors likely affect the availability of different resolution types. For example, self-help may be limited by disaster damage, mutual assistance between familiar people may depend on existing social capital, and that between unfamiliar people may require an explicit request or proposal [16, 17]. Finally, in addition to common background factors, we were interested in personality traits as individual factors. Understanding personality traits, or individual psycho–behavioral characteristics, may be a promising approach for educational or cultural investigations into improving resilience [7, 11].”

C3.2.5) I have not seen any literature review section!? I am pretty sure there are some studies to be covered.

R3.2.5) We believe our new second paragraph now makes the literature review of what has previously been studied on individual factors as follows (L57–72; c.f., R3.2.2):

“Published research on such individual factors has primarily addressed factors that contribute to negative consequences, that is, exacerbating factors. In this regard, the social science and mental health literatures have provided rather different images of victims. In the social science literature, the image of victims is rather vague in terms of the difficulties they encounter. Proposals of vulnerability factors [9-11] have been based on reviews of studies that describe vulnerability in association with various outcomes throughout the disaster stages, including casualty, physical or mental health, post-disaster socioeconomic states, and disaster preparedness. They commonly point out low socioeconomic status as a significant vulnerability factor. On the other hand, the image of the victim is clear in the mental health literature, primarily addressing poor mental health outcomes, such as post-traumatic stress disorders (PTSD). Although various individual factors have also been identified as risk factors, robust examination of disaster exposure and psychosocial resources (e.g., psychiatric pre-morbidity and personality) seems to be unique to the mental health literature [12-14]. In both literatures, the contributions of several common individual factors such as age, sex, and education are controversial.”

C3.3) Method:

C3.3.1a) Is the data collected from 2011 year or 3600 people who experienced 2011? If the answer is former, why did this study conducted so late? If answer is the latter one, when was this survey conducted?

C3.3.1b) How did the author(s) collect the survey questions? By email, by post service, or some 3rd party organization or data source that the author(s) use it for this study?

R3.3.1) The questionnaire battery was mailed to the participants in early December 2013 and returned by mid-January 2014. We clarified the period and sending methods of the survey in the revised Methods section as follows (L116–122):

“In early December 2013, a questionnaire battery was mailed to 3,600 residents who were randomly sampled from the electoral registers (and thus were aged 20 years or older) of tsunami-affected districts or temporary settlements in the four most populated coastal cities (Ishinomaki, Kesen-numa, Natori, and Sendai) in Miyagi Prefecture, where the damage caused by the earthquake and tsunami was most severe. In total, 1,412 questionnaires (39%) were anonymously completed and returned by mid-January 2014.”

C3.3.2) Why did the author(s) select 18 categories? Why not 15 or 20?

R3.3.2) It was not our intention to choose ‘18’ areas or items. We tried to cover as many difficulties as possible based on our own experience and knowledge, and as a result, we ended up with 18 items. We admit, however, that the original phrase "The following 18 items were selected:" makes it sound as if we intentionally chose 18 items. Therefore, we removed this number and modified it as follows (L133–139):

“The areas of difficulty in the aftermath of the earthquake and tsunami were selected to cover as many difficulties as possible based on our own experience and knowledge. The following items were selected: 1) eating, 2) cooking, 3) appropriate spare clothes, 4) room temperature, 5) sleeping, 6) access to a toilet, 7) washing one’s face, 8) bathing, 9) laundry, 10) information gathering, 11) transportation, 12) medical care for oneself, 13) medical care for one’s family, 14) psychological stress, 15) psychological care for one’s family, 16) noise, 17) stench, and 18) privacy.”

C3.3.3) What is the DV in this study? You observed the 18 categories on what?

R3.3.3) The dependent variable of our primary analysis was the resolution success of each difficulty type (i.e., obtained from the factor analysis of the 18 difficulty items) separately for the six solution types (self-help, request for help, help from family, help from an acquaintance, help through cooperation, and public assistance). That is, our primary analysis composed of 12 regression analyses with different dependent variables and the same set of explanatory variables. We admit that our original manuscript did not provide sufficiently clear view of the purpose and primary analysis. We believe that they are now clear thorough the clarification of the aim (L87–89; see R3.2.4) and method outline in revised Introduction as follows (L99–103):

“First, we explored the major types of difficulty among the 18 areas using a cluster analysis; characteristics of these difficulty types were examined in terms of frequency, relevant resolution types, and exacerbating factors. Then we explored the individual factors that facilitated the resolution of each difficulty type separately for the six solution types.”

C3.4) Analysis: 

C3.4.1) How come the study was performed with hierarchical regression? What are the motives for this? The reader should be prepared for this in earlier section(s).

R3.4.1) We used hierarchical regression to examine the contribution of personality factors after controlling for the effect of general background (i.e., damage and demographic) factors. It is now explicitly explained in the method outline of the revised Introduction as follows (L103–105):

“To examine contributing individual factors, a hierarchical regression analysis was used. We first identified contributing background factors and then asked whether any personality factor had additional explanatory power.”

C3.4.2) Some of the tables should be condensed or removed or reorganized. It seems too fluffy by having more than 15 tables! More like a report I would say rather than a scientific paper.

R3.4.2) We extensively reorganized the tables to improve the readability. Original Table 1 was reformatted to new Table 1 and original Tables 2 and 3 are now in the supplementary (S1 and Tables, respectively). Original Table 4 was divided into the degree of difficulty and resolution success (new Tables 2 and 3, respectively). Original Tables 5-18 are now all in the supplementary (S3-16 Tables, respectively).

C3.5) Discussion and conclusions

C3.5.1) Same comment applies for the discussion. Just discuss the main findings. It seems too long.

R3.5.1) Discussion was thoroughly reorganized and rewritten to improve the clarity. Particularly, three long paragraphs previously dedicated to extended (e.g., anthropological) discussion on each personality factor identified were now totally removed. Together with the clarification of the research purpose, we focused on our main finding and believe the revised Discussion is now much more readable. As a result, the length of our Discussion was drastically diminished (from 2,896 to 1,553 words).

C3.5.2) On the other hand, conclusion should be highlighted as it is the most important part of this study.d..

R3.5.2) Conclusions section was totally rewritten to highlight our most important findings as follows (L488–514): 

“Analyzing the data from the 2011 Great East Japan Earthquake and Tsunami Disaster, we explored individual background and personality factors that facilitate self-aid or mutual assistance in the immediate aftermath of a disaster considering different types of difficulty and resolution. We first identified two types of difficulties: general life difficulties, which were experienced by most victims, and medico-psychological difficulties, which were experienced to varying degrees across victims. Both types were exacerbated by disaster damages, whereas medico-psychological difficulties were uniquely affected by personality factors, with neuroticism and extraversion acting as exacerbating and mitigating factors, respectively. 

Then we examined the effect of individual factors on success in resolving the two types of difficulty through six types of resolution: self-resolution, four types of mutual assistance, and public assistance. In general, disaster damage hampered self-resolution and forced a reliance on cooperation or public assistance. Some demographic factors, such as being young and living in a three-generation household, facilitated resolution thorough the family. Importantly, several personality factors facilitated different types of resolution, primarily of general life difficulties: problem solving facilitated self-resolution, altruism or stubbornness facilitated resolution through request, leadership facilitated resolution through acquaintances, and emotion regulation facilitated resolution through public assistance. 

Our results demonstrate the importance of dissociating difficulties and resolution types when identifying facilitative factors for self-aid and mutual assistance in this disaster stage. The identified involvement of different individual factors, particularly personality factors, in different types of resolution may reflect the complexity of social dynamics as well as the psychological processes involved in survival in this context. The current findings contribute to the improvement of disaster risk mitigation through risk evaluation and the fostering of resilience among citizens by allowing separate consideration of different difficulty types and resolution processes.”

---

## [Decision Letter · Decision Letter 2]

27 Sep 2021

Self-help and mutual assistance in the aftermath of a tsunami: how individual factors contribute to resolving difficulties

PONE-D-20-29963R2

Dear Dr. Sugiura,

We’re pleased to inform you that your manuscript has been judged scientifically suitable for publication and will be formally accepted for publication once it meets all outstanding technical requirements.

Kind regards,

Dr. Md Nazirul Islam Sarker

Academic Editor

PLOS ONE

Additional Editor Comments (optional):

The author is requested to keep in touch with production team for further publication process.

Reviewers' comments:

Reviewer's Responses to Questions

**Comments to the Author**

1. If the authors have adequately addressed your comments raised in a previous round of review and you feel that this manuscript is now acceptable for publication, you may indicate that here to bypass the “Comments to the Author” section, enter your conflict of interest statement in the “Confidential to Editor” section, and submit your "Accept" recommendation.

Reviewer #1: All comments have been addressed

Reviewer #3: All comments have been addressed

2. Is the manuscript technically sound, and do the data support the conclusions?

Reviewer #1: Yes

Reviewer #3: Yes

3. Has the statistical analysis been performed appropriately and rigorously? 

Reviewer #1: Yes

Reviewer #3: Yes

4. Have the authors made all data underlying the findings in their manuscript fully available?

Reviewer #1: Yes

Reviewer #3: Yes

5. Is the manuscript presented in an intelligible fashion and written in standard English?

Reviewer #1: Yes

Reviewer #3: Yes

6. Review Comments to the Author

Reviewer #1: Section Re-Review comments

Abstract, title

and references

-In the 2nd revision, the author has addressed the issues like, revised the abstract properly, defined the study aims, data collection time and implication of the study accordingly.

Introduction/ Background

-The authors have revised the manuscript based on reviewer comments and suggestion.

-I expected to the should use different color to mark the revised sentence.

Methods -they addressed all the issues raised by reviewer

Results -The authors have concise this part according to the reviewer suggestion.

-they have addressed other issues marked by reviewer

Discussion and Conclusions

-the author has revised their manuscript based on reviewer comments.

-the authors read carefully all other comments and revised the manuscript accordingly.

-They have updated the reference section and followed the Journal guidelines.

Overall

I’ve read the author responses/revision and the 2nd time revised manuscript whether they have addressed the issues properly or not. Based on my understanding the authors have addressed all the comments and revised their manuscript appropriately.

Overall, the paper may consider for publication.

Reviewer #3: Thank you for addressing my concerns.

Except for the methodology section of clarifying the reasons for hierarchical regression and motives, I can see my comments are addressed.

7. PLOS authors have the option to publish the peer review history of their article (what does this mean?). If published, this will include your full peer review and any attached files.

Reviewer #1: **Yes: **Md. Nuralam Hossain

Reviewer #3: No

---

## [Editor Report · Acceptance letter]

29 Sep 2021

PONE-D-20-29963R2 

Self-help and mutual assistance in the aftermath of a tsunami: how individual factors contribute to resolving difficulties 

Dear Dr. Sugiura:

I'm pleased to inform you that your manuscript has been deemed suitable for publication in PLOS ONE. Congratulations! Your manuscript is now with our production department. 

Kind regards, 

on behalf of

Dr. Md Nazirul Islam Sarker 

Academic Editor

PLOS ONE